# The Effects of Iridin and Irigenin on Cancer: Comparison with Well-Known Isoflavones in Breast, Prostate, and Gastric Cancers

**DOI:** 10.3390/ijms26062390

**Published:** 2025-03-07

**Authors:** Yaeram Won, Hun-Hwan Kim, Se-Hyo Jeong, Pritam Bhagwan Bhosale, Abuyaseer Abusaliya, Jeong-Doo Heo, Je-Kyung Seong, Mee-Jung Ahn, Hye-Jung Kim, Gon-Sup Kim

**Affiliations:** 1Department of Pharmacology, Institute of Medical Sciences, College of Medicine, Gyeongsang National University, Jinju 52727, Republic of Korea; yrwon01@gmail.com (Y.W.); hyejungkim@gnu.ac.kr (H.-J.K.); 2Department of Convergence Medical Science, Institute of Health Sciences, College of Medicine, Gyeongsang National University, Jinju 52727, Republic of Korea; 3Research Institute of Life Science, College of Veterinary Medicine, Gyeongsang National University, Jinju 52828, Republic of Korea; shark159753@naver.com (H.-H.K.); tpgy123@gmail.com (S.-H.J.); shelake.pritam@gmail.com (P.B.B.); yaseerbiotech21@gmail.com (A.A.); 4Biological Resources Research Group, Gyeongnam Department of Environment Toxicology and Chemistry, Korea Institute of Toxicology, 17 Jegok-gil, Jinju 52834, Republic of Korea; jdher@kitox.re.kr; 5Laboratory of Developmental Biology and Goenomics, BK21 PLUS Program for Creative Veterinary Science Research, Research Institute for Veterinary Science, College of Veterinary Medicine, Seoul National University, Seoul 08826, Republic of Korea; snumouse@snu.ac.kr; 6Department of Animal Science, College of Life Science, Sangji University, Wonju 26339, Republic of Korea; meeahn@sangji.ac.kr

**Keywords:** daidzein, genistein, glycitein, irigenin, iridin, isoflavones, signaling pathway

## Abstract

Cancer, a worldwide problem and one of the leading causes of death due to uncontrolled cell proliferation, can be caused by various factors, such as genetic and environmental factors. Apoptosis is a programmed cell death mechanism that eliminates abnormal cells or renews cells. There are two main apoptotic pathways: intrinsic and extrinsic pathways. These pathways can be affected by various signaling pathways in cancer, such as the PI3K/AKT, MAPK, Wnt, and JAK/STAT pathways. Numerous approaches to cancer treatment have been studied, and among them, natural compounds have been actively researched. Flavonoids are natural compounds from fruits and vegetables and have been studied for their anti-cancer effects. Isoflavones, one of the subclasses of flavonoids, are usually found in soy food or legumes and are effective in several bioactive functions. The well-known isoflavones are genistein, daidzein, and glycitein. Irigenin and iridin can be extracted from the Iris family. Both irigenin and iridin are currently being studied for anti-inflammation, antioxidant, and anti-cancer by inducing apoptosis. In this review, we summarized five isoflavones, genistein, daidzein, glycitein, irigenin, and iridin and their effects on three different cancers: breast cancer, prostate cancer, and gastric cancer.

## 1. Introduction

Cancer is one of the leading causes of death worldwide due to abnormal cell proliferation [1,2]. Cancer can be caused by genetic and environmental factors, including smoking, drinking, obesity, and lifestyle habits [3]. According to 2022 statistics from the International Agency for Research on Cancer (IARC), breast cancer is the second, prostate cancer is the fourth, and gastric cancer is the fifth most diagnosed cancer worldwide. Among these cancers, breast cancer is the third, and gastric cancer is the fifth most common cause of death [4].

Apoptosis is programmed cell death designed to eliminate cells that are damaged or no longer needed and play an important role in cancer cells [5]. The balance between cell proliferation and programmed cell death is critical for survival, but when the balance is disturbed, defects in apoptosis occur in the cancer cell [6]. Apoptosis can be divided into two pathways: extrinsic pathway and intrinsic pathway [7]. The extrinsic pathway is regulated through external stimuli when death ligands bind to their death receptors, while the intrinsic pathway is mediated through mitochondria when ROS levels are high, or DNA is damaged [8]. Many studies have shown that natural compounds can induce apoptosis in cancer cells [9]. Especially, flavonoids, one of the secondary metabolites of plants, showed anti-cancer effects [10].

Isoflavones are a natural phenolic compound mostly found in legumes and are one of the subclasses of flavonoids [11]. Isoflavones are called phytoestrogen due to their structural similarity to estrogen, and they perform a role similar to that of estrogen [12]. Recent studies have reported an anti-obesity effect, a mechanism to lower blood sugar levels, and benefits for osteoporosis [13]. There are two types of isoflavones: glycoside form and aglycone form. The most well-known aglycone forms are genistein, daidzein, and glycitein, and their glycoside forms are genistin, daidzin, and glycitin [14]. Currently, isoflavones are being studied for their anti-cancer effects and their ability to induce cell apoptosis in hormone-related breast and prostate cancers [12].

Iridin, one of the isoflavones, is known for its potential antioxidant, anti-cancer, and anti-diabetic effects [15]. One study shows that iridin can induce cell apoptosis and reduce inflammation [16,17]. Irigenin, the aglycone of iridin, is an O-methylated isoflavone, and some studies indicate that it has anti-cancer and anti-inflammatory effects [18]. However, the metabolism of iridin and irigenin has not yet been fully revealed, and additional research is needed.

In this review, we first explain the apoptosis and signaling pathway. Then, we summarize five types of isoflavones genistein, daidzein, glycitein, irigenin, and iridin and their effects on breast cancer, prostate cancer, and gastric cancer. For a comprehensive literature overview, we used several search engines such as PubMed and Google Scholar using the terms ‘daidzein’, ‘genistein’, ‘glycitein’, ‘irigenin’, ‘iridin’, ‘isoflavones’, and ‘signaling pathway’.

## 2. Apoptosis Pathway

The intrinsic pathway induces apoptosis through internal stress, such as biochemical stress and lack of Growth Factors. The intrinsic pathway, also called the mitochondrial intrinsic pathway, is mediated through mitochondria [19]. The Bcl-2 family regulates the intrinsic pathway and can be classified into pro- and anti-apoptosis. The type of pro-apoptosis includes Bax and Bak, while anti-apoptosis will be Bcl-xL, Bcl-w, and Bcl-B [20]. When internal signals such as DNA damage or oxidative stress are sensed, Bax and Bak regulate the mitochondria to release cytochrome C from MOMP (Mitochondrial outer membrane permeabilization) to the cytoplasm [21]. When cytochrome C is released to the cytoplasm, it binds with Apaf-1 and pro-caspase-9 to form apoptosome, which is a cytoplasmic death-inducing complex involved in apoptosis [7]. The apoptosome activates caspase-9, which activates caspase-3 to induce apoptosis (Figure 1) [19]. One study has shown that both genistein and daidzein initiated the intrinsic pathway to induce apoptosis. Both upregulated the expression of BAX and downregulated the expression of Bcl-2 to release cytochrome C from mitochondria to cytosol in order to induce apoptosis in tumors [22].

The extrinsic pathway, which is also called the death receptor pathway, induces apoptosis through external stimuli when the death receptor binds to its ligands on the cell surface (Figure 1) [7]. Death receptors are members of the tumor necrosis factor (TNF) receptor superfamily, characterized by extracellular cysteine-rich regions and a cytoplasmic region known as the death domain (DD). The DD enables the receptors to initiate and transmit cytotoxic signals upon binding with their cognate death ligands, ultimately leading to apoptosis [23]. The key types of the TNF receptors involved in this pathway are FasR, TNF-R1, and TNF-related apoptosis-inducing ligand receptors (TRAIL-Rs), while the key types of ligands are FasL, TNF- α, and TRAIL [7]. When FasL binds to its receptor FasR, it forms a Fas-associated death domain (FADD), and when TNF- α binds to its receptor TNF-R, it forms a TNF receptor-associated death domain (TRADD). FADD and TRADD are called adapter proteins, which form a complex called the death-inducing signaling complex (DISC), which plays an important role in activating pro-caspase-8. When pro-caspase-8 is activated, it activates caspase-8 and -3 to induce apoptosis (Figure 1) [24]. Iridin has been reported to induce an extrinsic pathway by exhibiting the Fas-mediated apoptotic cell death in AGS cells by regulating the PI3K/AKT signaling pathway [16].

## 3. Various Signaling Pathways That Are Involved in Apoptosis in Cancer

There are various signaling pathways involved in apoptosis in cancer. Five different isoflavones can either enhance or inhibit the signaling pathway to induce apoptosis through several signaling pathways.

The PI3K/AKT pathway plays an essential role in cell metabolism, cell survival, and cell proliferation [25]. In Figure 2, phosphatidylinositol-3 kinase (PI3K) consists of catalytic subunit p110, adapter/regulatory subunit p85, and PI3K phosphorylate phosphatidylinositol-4, 5-bisphosphate (PIP2) to phosphatidylinositol-3, 4, 5-trisphosphate (PIP3) [26]. PIP3 provides a docking site to 3-phosphoinositide-dependent kinase (PDK1) and mTORC2 and phosphorylates AKT serine/threonine kinase [26]. Activated AKT phosphorylates the pro-apoptotic factor BAD and caspase-9 to inhibit apoptosis [27]. In cancer, the PI3K/AKT pathway is hyper-activated. In breast cancer, for example, PI3K is hyper-activated due to the loss of PI3K inhibitory functions and mutation in tumor suppressor genes [28]. As I mentioned above, iridin inhibits PI3K/AKT signaling by reducing the expression of p-AKT and p-PI3K, resulting in the suppression of cell proliferation, induction of G2/M phase cell cycle arrest, and exhibition of Fas-mediated extrinsic apoptotic cell death [16].

Mitogen-Activated Protein Kinase (MAPK) plays an essential role in cell proliferation, cell division, cell aging, and apoptosis [29]. MAPK consists of three main kinases: MAPK, MAPKK, and MAPKKK. MAPK is phosphorylated by MAPKK, and MAPKK is phosphorylated by MAPKKK [30]. MAPK has six subclasses in mammals, which are extracellular signal-regulated kinase (ERK) 1/2, ERK3/4, ERK5, ERK7/8, Jun N-terminal kinase (JNK) 1/2/3, and p38 MAPK [31]. The Erk signaling pathway and p38 pathway are involved in anti-apoptosis through cell proliferation, survival, and differentiation [30]. The JNK pathway signal targets the mitochondria for pro-apoptosis to induce apoptosis (Figure 2) [31]. In breast cancer, for example, genistein triggers G2/M cell cycle arrest in MDA-MB-231 via RAS/MAPK/activator protein-1 and downregulates CDK1, cyclin B1, and CDC25C [32]. Additionally, the treatment of genistein inhibits the phosphorylation of p38, p42/44, and p-JNK [33]. Additionally, irigenin downregulates the ERK/MAPK signaling pathway by reducing the expression of p-P38 and p-ERK in irigenin-treated Coca-2 cells [34].

The Wnt pathway regulates cell proliferation, cell division, and apoptosis [35]. Wnt is a ligand that binds to a cell surface receptor called Frizzled to activate (Figure 2) [36]. When Wnt is absent, cytoplasmic degradation complex, which consists of axin, adenomatous polyposis coli (APC), glycogen synthase kinase 3 (GSK-3), and β-catenin, is phosphorylated by GSK-3. As a result, β-catenin is maintained at a low concentration by axin and APC and degraded in the proteasome [37]. When Wnt binds to its receptor, CK1 and GSK-3 phosphorylate β-catenin and β-catenin translocate to the nucleus [35]. β-catenin binds to the T-cell factor (TCF) and activates the gene to be expressed. Overexpression of the Wnt/β-catenin pathway has been studied as a leading factor in many cancer types. The reasons for overexpression factors are mutations in the β-catenin gene, abnormalities in the β-catenin destruction complex, APC mutations, overexpression of Wnt ligands, loss of inhibition, or decreased activity of regulatory pathways [38,39]. 

The Janus Kinase/Signal Transducers and Activators of Transcription (JAK/STAT) pathway plays an essential role in cell proliferation, cell division, and apoptosis (Figure 2) [40]. The JAK/STAT pathway consists of cellular receptors, JAK proteins, and STAT proteins. The JAK protein consists of JAK1, JAK2, JAK3, and TYK2, while the STAT protein consists of STAT 1, STAT 2, STAT 3, STAT 4, STAT 5A, STAT 5B, and STAT 6 [41]. When ligands such as cytokine, Growth Hormone (GH), and Growth Factor (GF) bind to its receptor, JAK will activate to phosphorylate STAT. Activated STAT will form a dimer and translocate into the nucleus. STAT will bind to a specific DNA sequence and regulate the transcription of the target genes [42]. The activated JAK/STAT pathway regulates cancer survival and transition. For example, one study has demonstrated that the JAK/STAT signaling pathway is activated in breast cancer by the binding of the IL-6 family of cytokines to their receptors [43]. One study has proven that iridin binds to the active site of PKM2 to inhibit the expression of PKM2, which can downregulate the JAK/STAT signaling pathway [17].

## 4. Anti-Cancer Effects of Various Isoflavones

### 4.1. Isoflavones Structure and Their Role in Human Health

Isoflavones (Figure 3F), a subclass of flavonoids, are usually found in soy food and are called phytoestrogen due to their structural similarity with the hormone estrogen [44]. Phytoestrogens are compounds that have estrogen-like effects on humans and can bind to estrogen receptors [45]. Briefly, flavonoids are a secondary metabolism substance derived from plants and fruits, a natural compound, and a subclass of polyphenols [46]. Flavonoids have a C6-C3-C6 carbon structure (Figure 3A), consisting of two benzene rings and a 3-carbon chain in the middle, except for chalcone and stilbene, which have a C6-C2-C6 carbon structure (Figure 3B,C). Flavonoids can be divided into 12 subclasses, which are shown in Figure 3B–M [47]. There are two main types of isoflavones: glycoside and aglycone. The main glycosides of isoflavone are genistin, daidzin, and glycitin, while the main aglycones are genistein, daidzein, and glycitein (Figure 4) [48]. Usually, natural isoflavones are in the form of glycoside rather than aglycone [11,13]. When ingested, the glycoside form is hydrolyzed into aglycone and can be absorbed into our body [11,49]. Recently, many researchers have given attention to isoflavones because of their beneficial effects on human health. Some studies suggest that isoflavones show antioxidant and anti-inflammatory properties [50,51]. These studies have led to increased interest in the potential anti-cancer effects of isoflavones, and active research has been conducted, especially on breast and prostate cancer, as they are phytoestrogens [12,52].

### 4.2. Various Isoflavones and Their Effects on Anti-Cancer

#### 4.2.1. Genistein and Its Effects on Breast, Prostate, and Gastric Cancers

Genistein, one of the predominant isoflavones, is mostly found in and extracted from soy products, such as soybeans and soy milk. Its chemical structure is 4′,5,7-trihydroxyisoflavone (C15H10O5) [53]. Genistin is the glycoside form of genistein substituted by 7-ο-β-D-glucoside from genistein [54]. Several studies suggest that genistein can inhibit the cell cycle and suppress metastasis and angiogenesis [55,56]. These studies indicate genistein and genistin have potential anti-cancer effects on breast cancer, prostate cancer, and gastric cancer [57]. In MCF-7 breast cancer, treatment with 50 to 100 µM of genistein arrested the G2/M cycle [58,59]. Also, genistein treatment at 10, 25, and 50 µM showed a decrease in cell proliferation, caused apoptosis by increasing the expression of BAX, and decreased cell invasion and migration in both MDA-MB-231 and MCF-7 breast cancer [60]. In one study, genistein was treated in five different breast cancer cells, which are MDA-MB-231, MDA-MB-468, MCF-7, T-47D, and MCF-10A as control. Cells were treated with genistein at concentrations ranging from 10 to 200 µM for 24 h and 48 h. MCF-7 had the lowest percentage of viability, and T-47D had the highest viability [59,61]. In PC3 prostate cancer cells, 30, 50, and 70 µM of genistein prevented migration, inhibited proliferation by reducing p38 MAPK, and induced apoptosis by enhancing caspase-9 expression [62]. One study was conducted using three different prostate cancer cell lines: LNCaP, DU 145, and PC-3. Each prostate cancer was treated with 5α-Dihydrotestosterone to induce the expression of the prostate androgen-regulated transcript-1 (PART-1), and only LNCaP expressed PART-1. Genistein was treated at different concentrations for 24 h. The result revealed that 12.5, 25, 50, and 100 µmol/L inhibited the PART-1 expression, which regulated the transcriptome of prostate cancer [63]. The proliferation of LNCaP and DU145 cell lines was assessed with genistein at concentrations of 0, 10, 25, and 50 µM, showing a significant decrease in both cell lines [64]. In BGC-823 human gastric cancer cells, genistein inhibited the activation of NK-ĸB and decreased the concentration of COX-2, inhibiting cell proliferation and inducing apoptosis. Additionally, in SGC-7901 and BGC-823 cells, genistein arrested the g2/M cell cycle and hyper-activated Phosphatase and Tensin Homolog (PTEN), leading to AKT deactivation. As a result, it decreased the phosphorylation of Ser642 and Wee1 and phosphorylated CDC2/CDK1 (Table 1) [65,66,67].

#### 4.2.2. Daidzein and Its Effects on Breast, Prostate, and Gastric Cancers

Daidzein is also one of the abundant isoflavones found and extracted from soybeans. Its chemical structure is 7 [7-hydroxy-3-(4-hydroxyphenyl)-4-1benzopyran-4-one] [80]. Currently, it is being studied for its potential anti-cancer effects on breast cancer, prostate cancer, and gastric cancer [81]. In MCF-7 breast cancer, treatment with 25 to 100 µM of daidzein activated caspase-9, inducing apoptosis of the intrinsic pathway [69,70]. Another study showed cell growth inhibition and activation of caspase 3/7, inducing apoptosis in MCF-7 cells [69]. Additionally, daidzein was tested on two different breast cancer cell lines, MCF-7 and MDA-MB-231. The result showed that, in a dose-dependent manner from 10 to 200 µM, both cancer cell lines’ viability percentage was low. In both breast cancers, daidzein inhibited cell survival by targeting the PI3K/Akt pathway [70]. In PC3 prostate cancer, daidzein treatment induces apoptosis by increasing Bax expression and lowering inhibitors of apoptosis (IAPs) [71]. One study was conducted using LNCaP, DU 145, and PC-3 to see whether daidzein inhibits the expression of PART-1 in LNCaP. Similar to genistein, daidzein also inhibited the expression of PART-1, but the effect was less than genistein [82]. Also, the proliferation of LNCaP and DU145 cell lines treated with daidzein at concentrations of 0, 10, 25, and 50 µM revealed a significant reduction in both cell lines [64]. In BGC-823 gastric cancer cells, treatment with 20 to 80 µM of daidzein inhibited cell proliferation. Daidzein decreased the concentration of Bcl-2, increased Bax concentration, and regulated caspase-9 and caspase-3 to induce apoptosis of the intrinsic pathway (Table 1) [72].

#### 4.2.3. Glycitein and Its Effects on Breast, Prostate, and Gastric Cancers

Glycitein is the third most abundant in isoflavone, mostly extracted from soybean, and its structure is 4′, 7-dihydroxy-6-methoxyisoflavone [83]. Glycitein also has potential anti-cancer effects on breast cancer and gastric cancer but not on prostate cancer [73,84]. In breast cancer SKBR-3 cells, glycitein damages the cell surface and increases the permeability to suggest the anti-cancer effect [32,73]. Also, glycitein showed strong anti-proliferative effects on MDA-MB-231 [74]. Additionally, glycitein significantly decreased p-STAT3 and downregulated p-Akt, pmTOR, and p-p38 in MCF-7 to induce apoptosis [75]. Glycitein induces ERK 1/2 activity in nontumorigenic RWPE-1 prostate epithelial cells, but currently, there are not enough studies on prostate cancer [85,86]. In gastric cancer AGS cells, an experiment was conducted to assess the effects of glycitein treatment at 30 µM for 3, 6, 12, and 24 h. As a result, glycitein arrested the G0/G1 phase cycle and inhibited the STAT3/NF-kB signaling pathway. Also, glycitein activated the caspase cascade and produced ROS to activate MAPK to induce apoptosis (Table 1) [76].

#### 4.2.4. Irigenin/Iridin and Its Effects on Breast, Prostate, and Gastric Cancers

Irigenin is usually extracted from the Iris family and is the aglycone form and major metabolite of iridin [87]. Irigenin is an O-methylated isoflavone (Figure 5A), and several studies have shown its effects on cell signaling. For example, one study suggested that irigenin arrested the cell cycle by increasing the proportion of cells in the G2/M phase, decreasing cyclin B1, and suppressing migration and the YAP/β-catenin signaling pathway in glioblastoma cells to induce apoptosis [88]. Research has proven that irigenin inactivates the MAPK signaling pathway in LPS-treated mice by regulating the inflammatory response, cytokine production, and cell death in acute lung injury (Figure 6) [89]. Also, irigenin inhibits the expression of YAP, which reduces the expression of β-catenin to inhibit cell proliferation (Figure 6) [88]. Based on some studies related to cell signaling, interest in research on irigenin as an anti-cancer arose, especially on prostate and gastric cancers, but fewer studies are conducted on breast cancer [18]. In prostate cancer cells, RWPE-1, LNCaP, and PC3, treatment with 50 to 100 µM of irigenin inhibited cell proliferation by arresting the cell cycle in the G1 phase and inhibiting p21 and p27 protein expression [18]. One study shows that irigenin did not show any anti-cancer effects in MCF-7 and T-47D breast cancer cells [18]. However, irigenin inhibits the production of NO and PGE2 induced by LPS [90]. Another study examined the effects of irigenin on TRAIL-resistant gastric cancer cells. The use of irigenin independently did not show any effects on the cancer. However, the use of the combination of TRAIL and irigenin activated caspase-8/9/3 and PARP, hyper-expressed FADD, DR5, and BAX, and inhibited the expression of cFLIB, Bcl-2, and Survivin (Table 1) [79]. 

Iridin is also found in the Iris family, Belamcanda chinensis, Iris kumaonesis, and Iris florentina [16]. The chemical structure of iridin is the 7-glucoside of irigenin, mainly classified as aglycoside of irigenin (Figure 5B), and according to some studies, iridin shows anti-cancer, antioxidant, and anti-inflammation effects [17]. In the human gastric cancer cell line AGS and HaCaT cells, treatment with 12.5 to 200 µM of iridin induced apoptosis by inducing caspase-8 and caspase-3 in the extrinsic pathway. Also, in the PI3K/AKT pathway, iridin decreased the phosphorylation of PI3K and AKT to regulate apoptosis in AGS cells (Table 1) (Figure 6) [16].

## 5. Discussion

In this review, we mainly summarized five isoflavones’ effects on breast, prostate, and gastric cancers. Phytoestrogens found in soy products have led to numerous studies on human health, with isoflavones being one of the main phytoestrogen compounds actively studied in relation to breast, prostate, and gastric cancers [44].

One of the causes of cancer is inflammation, and isoflavones have been proven to exhibit anti-inflammatory effects invitro. One study was conducted to see the effect of black soybean extract (BSE), genistein, and daidzein on PGE-20, TNF-α, and IL-1β levels in LPS-induced RAW 264.7 cells using concentrations of 40 and 200 µg/mL, respectively. All three levels were decreased at 40 µg/mL of genistein and daidzein treatment compared to the control, showing the highest inhibitory activity over positive control [91]. Another study pretreated BV2 cells with genistein using concentrations of 25 and 50 µM for 1 h before LPS treatment to measure nitrite and PGE_2_ levels. Results showed that 50 µM significantly decreased the level of nitrite and PGE_2_ [92]. For daidzein, a study was conducted to examine IL-6 and TNF-α levels to investigate the anti-inflammatory activity of daidzin and daidzein at concentrations of 50 and 100 µM in LPS-induced RAW264.7 cells. Both daidzin and daidzein significantly reduced IL-6 release at both concentrations, but there was no difference in TNF-α release [93]. BV2 cells were pretreated with glycitein (5, 20, and 50 µM) for 30 min before LPS stimulation, and NO, TNF- α, and IL-1 β levels were measured 24 h later. The results showed that these concentrations significantly inhibited NO, TNF- α, and IL-1 β productions, but limited studies have been conducted due to the low abundance of glycitein in soy products [94,95]. Based on these studies, isoflavones show anti-inflammatory effects in vitro, leading to further investigation of their effects in vivo.

Several in vivo studies support that isoflavones can directly affect breast, prostate, and gastric cancers. In in vivo breast cancer studies, xenograft tumor growth of MCF-7 and MDA-MB-231 cells in nude mice was inhibited by genistein [96]. Additionally, 4T1 cells, which is an animal model for stage *IV* human breast cancer, were injected into BALB/c mice and treated with genistein (200 mg/kg) and centchroman (10 mg/kg), an estrogen receptor modulator, thrice a week for 3 weeks. Treatment of both genistein and centchroman significantly reduced the size of the volume compared to control and treatment of genistein or centchroman alone [68]. In another study, 4T1 breast cancer cells were injected into BALB/c mice, which were divided into four groups and treated with different intensities of exercise (0, 6, 10, and 15 m/min) and doses of daidzein (0, 45, 75, and 145 mg/kg). The result showed that regular exercise combined with daidzein significantly reduced tumor volume and size [97]. For prostate cancer in vivo studies, in one study, BIO 300, which is a proprietary nanosuspension of synthetic genistein, was shown to synergize with radiation, delaying tumor growth and extending survival in the xenograft model [98]. Human prostate cancer cells 22RV1 and DU145 were injected into nude mice and treated with genistein (10 mg/kg) and docetaxel (10 mg/kg), either together or separately. Results showed that treatment with only genistein and a combination of genistein and docetaxel significantly decreased the tumor volume [57]. Another study was conducted using a combination treatment of genistein and AG1042 with X-irradiation in a xenograft model of the prostate cancer cell lines PC3 and DU145. The results showed that the combination treatment significantly reduced tumor volume compared to individual treatment with X-irradiation [99]. Daidzein treatment showed roughly a 50% decrease in tumor weight and an almost 80% decrease with the combination of daidzein and genistein treatment in nude mice injected with PC-3 prostate cancer cells [100]. These in vivo studies prompted further investigation into their effects on human health.

For clinical trials, most of the studies are related to metabolism, such as decreasing gestational diabetes mellitus occurrence in pregnant women, reducing LDL cholesterol, increasing bone health, improving hot flashes, reducing CVD risk, enhancing antiaging effects, and promoting cognitive function [80,101,102]. Currently, there are several clinical trials of daidzein. Patients aged 65 years or older who took 60 mg of isoflavones, mainly containing daidzein, per day for 12 months showed low prostate cancer incidence [80,103]. Another clinical trial showed that patients who took 47 mg of isoflavones containing genistein and daidzein three times a day for 12 months had reduced prostate specific antigen (PSA) levels in prostate cancer [80,104]. These clinical trial studies did not show a clear impact on breast, prostate, and gastric cancers, but they were able to demonstrate effects on human health.

Isoflavones can also be extracted from plants within the Iris family, specifically compounds such as iridin and irigenin. These compounds have attracted attention because of their potential health benefits. Few studies indicate that iridin and irigenin exhibit promising effects in various cancer cell lines, including breast, prostate, and gastric cancers. Additionally, these compounds have shown anti-inflammatory and antioxidant effects in in vitro studies [18]. RAW 264.7 cells were pretreated with iridin (12.5, 25, 50 µM) for 2 h and then with LPS for 16 h. Compared to the control group, TNF- α, IL-1 β, MCP-1, and NO levels were significantly decreased in a dose-dependent manner [17]. However, the current body of research on the impact of iridin and irigenin in invitro studies, in vivo studies, and human health remains limited. More comprehensive studies are needed to better understand their efficacy, safety, and potential therapeutic application in human health.

To advance our understanding of isoflavones, especially on iridin and irigenin, research should prioritize longitudinal studies with standardized methodologies and diverse populations to apply to human health. Investigating the effects of other dietary foods on various physiological pathways could provide a more comprehensive view of our health benefits. Additionally, exploring the potential of personalized nutrition approaches based on individual metabolism might enhance the efficacy of isoflavones.

In summary, each isoflavone presents a distinct and promising natural alternative treatment to induce apoptosis in cancer treatment [105]. However, due to the limited studies, their application is constrained by variability in bioavailability. Therefore, more detailed examinations of isoflavones, especially in iridin and irigenin, can provide therapeutic potential and safety across diverse populations and clinical treatments in diverse cancers.

## 6. Conclusions

This review is about the potential anti-cancer effect of five isoflavones: genistein, daidzein, glycitein, irigenin, and iridin in breast cancer, prostate cancer, and gastric cancer. Iridin and irigenin, in particular, have not been studied much and need to be further investigated, and this review will help to guide further research. Additionally, this review includes two main pathways of apoptosis and various signaling pathways of apoptosis. Lastly, we hope this review will be helpful in researching the anti-cancer effect of isoflavone.

## Figures and Tables

**Figure 1 ijms-26-02390-f001:**
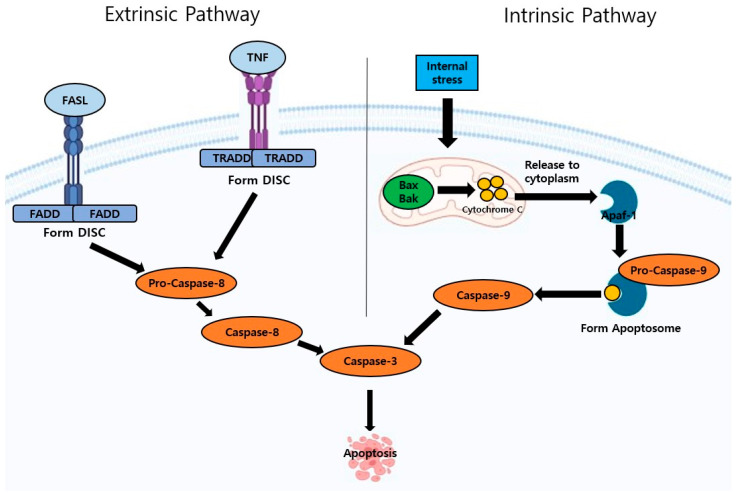
Extrinsic and intrinsic pathways.

**Figure 2 ijms-26-02390-f002:**
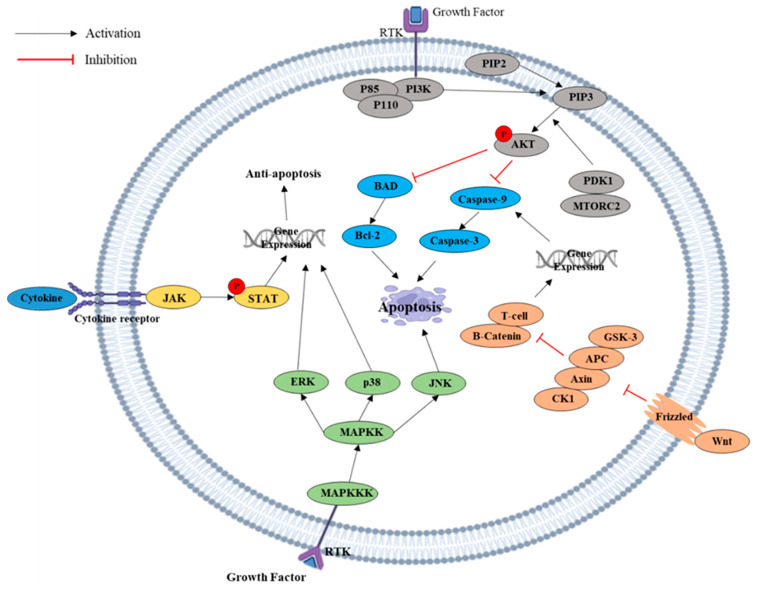
Various apoptotic signaling pathways in cancer.

**Figure 3 ijms-26-02390-f003:**
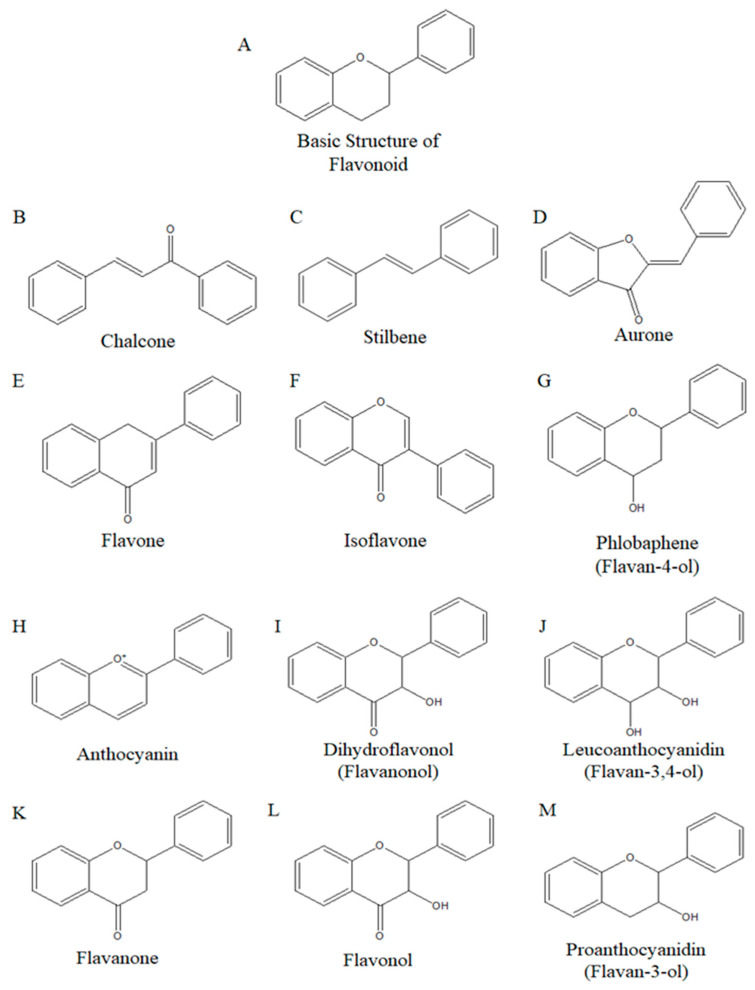
General structure of 12 subclasses of flavonoid. (**A**) basic structure of flavonoid, (**B**) chalcone, (**C**) stilbene, (**D**) aurone, (**E**) flavone, (**F**) isoflavone, (**G**) phlobaphene (Flavan-4-ol), (**H**) anthocyanin, (**I**) dihydroflavonol (Flavanonol), (**J**) leucoanthocyanidin (flavan-3,4-ol), (**K**) flavanone, (**L**) flavonol (**M**) proanthocyanidin (flavan-3-ol).

**Figure 4 ijms-26-02390-f004:**
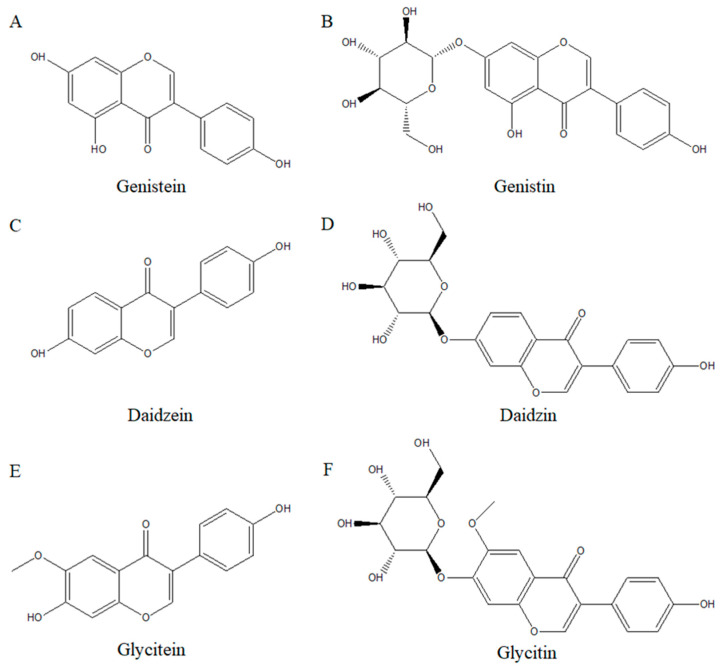
Isoflavone chemical structures in their three main aglycone forms: (**A**) genistein, (**C**) daidzein, and (**E**) glycitein; and three main glycosides: (**B**) genistin, (**D**) daidzin, and (**F**) glycitin.

**Figure 5 ijms-26-02390-f005:**
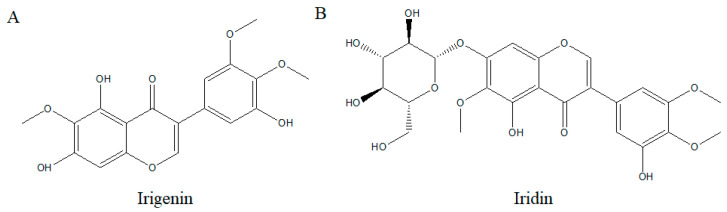
Chemical structure of (**A**) irigenin and (**B**) iridin.

**Figure 6 ijms-26-02390-f006:**
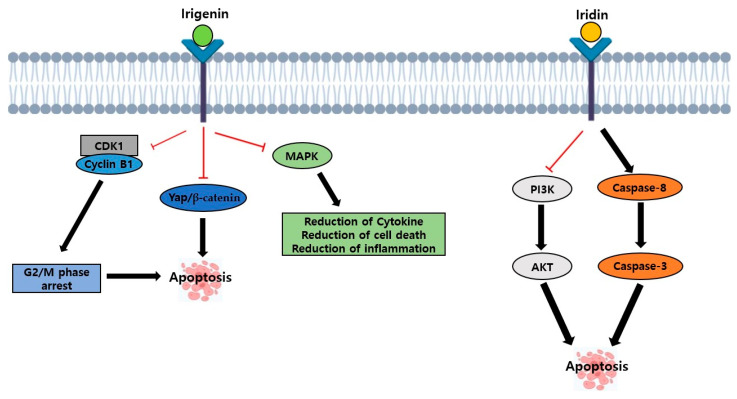
The effects of irigenin and iridin.

**Table 1 ijms-26-02390-t001:** Genistein, daidzein, glycitein, irigenin, and iridin effects on different cancer cell lines.

Genistein				
Cancer	Cell Line	Treatment	Effect	Reference
Breast cancer	MCF-7	50, 100 µM	Arrested the growth at G2/M phase	[59]
	MCF-7 MDA-MB-231	10, 25, and 50 µM	Decreased cell proliferation Increased BAX expression Decreased cell invasion/migration	[60]
	MDA-MB-231 MDA-MB-468 MCF-7 T-47D MCF-10A	10 to 200 µM For 24/48 h	Viability of all cell line low	[61]
	MCF-7	150 µM	Downregulated Bcl-2	[68]
Prostate cancer	PC3	30, 50, 70 µM	Increased caspase-3 Inhibited p38 MAPK	[62]
	LNCaP	12.5, 25, 50, and 100 µmol/L	Inhibited PART-1 expression	[63]
	LNCaP DU145	0, 10, 25, 50 µM	Decreased cell proliferation	[64]
Gastric cancer	BCG-823	20–80 µM	Inactivated AKT by upregulating PTEN	[65,67]
	SGC-7901 BGC-823	10, 20, 40, 80 µM	Arrested the growth at G2/M phase	[66,67]
**Daidzein**				
**Cancer**	**Cell Line**	**Treatment**	**Effect**	
Breast cancer	MCF-7	25, 50, 100 µM	Activated caspase-9	[69,70]
	MCF-7	50 µM	Inhibited cell proliferation Activated caspase-3, -7	[69]
	MCF-7 MDA-MB-231	10 to 200 µM	Low viability Inhibited PI3K/AKT pathway	[70]
Prostate cancer	PC3	50 µM	Increased Bax Decreased IAP	[71]
	LNCaP	12.5, 25, 50, and 100 µmol/L	Inhibited PART-1 expression	[63]
	LNCaP DU145	0, 10, 25, 50 µM	Decreased cell proliferation	[64]
Gastric cancer	BGC-823	20, 80 µM	Regulated caspase-3, 9 Decreased Bcl-2 Increased Bax	[72]
**Glycitein**				
**Cancer**	**Cell Line**	**Treatment**	**Effect**	
Breast cancer	SKBR-3	5, 10, 20 µM	Increased membrane permeability	[32,73]
	MDA-MB-231	0–200 µM	Inhibited proliferation	[74]
	MCF-7	0–200 µM	Downregulated phosphorylated STAT3, AKT, mTOR, and p38	[75]
Gastric cancer	AGS cell	30 µM	Arrested the growth at G0/G1 phase Inhibited STAT3/NF- kB pathway Activated caspase cascade Activated MAPK pathway	[76]
**Irigenin/Iridin**				
**Cancer**	**Cell Line**	**Treatment**	**Effect**	
Breast cancer	MCF-7 T-47D	10 nM to 100 μM	No effects	[18,77]
Prostate cancer	RWPE-1 LNCaP PC3 Cell	50, 100 µM	Arrested G1 phase Inhibited p21, p27 protein	[78]
Gastric cancer	TRAIL-resistant gastric cancer cell AGS HaCaT	12.5, 200 µM	Inhibited PI3K/AKT pathway Decreased caspase-3 and 8	[16,79]

## Data Availability

All data and analysis are available within the manuscript, or upon request to the corresponding authors.

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
