# Peer review of "The Effects of Iridin and Irigenin on Cancer: Comparison with Well-Known Isoflavones in Breast, Prostate, and Gastric Cancers"

_ijms, 2025, doi:10.3390/ijms26062390_

Round 1
Reviewer 1 Report (New Reviewer)
Comments and Suggestions for Authors
The review paper should be prepared using primarily the most recent references. The authors should update the reference list with new papers and revise the manuscript according to the findings in these new papers.
Author Response
For reviewer
Thank you for your attentive comments.
Comments and Suggestions for Authors
The review paper should be prepared using primarily the most recent references. The authors should update the reference list with new papers and revise the manuscript according to the findings in these new papers.
- I updated most of the references to the recent versions. Some of the references are old because they are experiment-based articles.

Reviewer 2 Report (Previous Reviewer 1)
Comments and Suggestions for Authors
In the current review, the authors presented five isoflavones - genistein, daidzein, glycitein, irigenin, and iridin and their effects on three different cancers - breast cancer, prostate cancer, and gastric cancer.
Some comments:
-In my opinion in the title you must specify that you wrote about breast, prostate and gastric cancers.
- Lines 91-96: the information it/s not about cancer.
-Lines 96-97, you wrote that the studies presented at lines 91-96 “lead the interest to have potential effects in anti-cancer, and active research has been conducted especially on breast and prostate cancer”. Why especially on breast and prostate cancer? Which is the connection between their anti-atherosclerotic activity and cancer and between their anti-osteoporotic activity and cancer? Please explain.
-Line 226 – you wrote about liver cancer.
-Lines 228-229, you wrote that “irigenin protect rats retina that are damaged from UVB”. Which is the connection with your review?
-Lines 321-323, you wrote “Obesity, diabetes, and inflammation increase the risk of breast, prostate, and gastric cancer and currently isoflavones are proven to have anti-obesity, anti-diabetes, and anti-inflammatory effects in-vitro”. Your review is about cancer or about preventing cancer?
Author Response
For reviewer, Thank you for your attentive comments.
- In my opinion in the title, you must specify that you wrote about breast, prostate and gastric cancers.
- I revised the title.
- Lines 91-96: the information is not about cancer.
- Yes, but I wrote this part because I wanted to explain that isoflavones are commonly found in soy beans, exist in two forms, and form changes when we consume them.
- Lines 96-97, you wrote that the studies presented at lines 91-96 “lead the interest to have potential effects in anti-cancer, and active research has been conducted especially on breast and prostate cancer”. Why especially on breast and prostate cancer? Which is the connection between their anti-atherosclerotic activity and cancer and between their anti-osteoporotic activity and cancer? Please explain.
- I added more detailed information about the question ‘why especially on breast and prostate cancer.’
- For anti-atherosclerotic and anti-osteoporotic activity, I wrote it because isoflavones have been studied in relation to human health and led to interest in their potential anti-cancer effects. However, going through your comments and reading my manuscript, it seems not to be related to cancer. So, I deleted those two activities and added anti-oxidation and anti-inflammation
- Line 226 – you wrote about liver cancer.
- Lines 228-229, you wrote that “irigenin protect rats’ retina that are damaged from UVB”. Which is the connection with your review?
- To answer the two comments, I wrote these two parts to give examples about irigenin effects in intrinsic pathway. However, because my topic is about isoflavones effects in breast, prostate, and gastric cancer, it seems not to be related to my topic. So, I deleted these two parts.
- Lines 321-323, you wrote “Obesity, diabetes, and inflammation increase the risk of breast, prostate, and gastric cancer and currently isoflavones are proven to have anti-obesity, anti-diabetes, and anti-inflammatory effects in-vitro”. Your review is about cancer or about preventing cancer?
- My review is to summarize and comprehend the effects of isoflavones on three types of cancer based on existing research. I wrote obesity, diabetes, and inflammation part to discuss on other mechanisms of action of isoflavones because these three play a role in cancer development and anticancer activity.

Round 2
Reviewer 1 Report (New Reviewer)
Comments and Suggestions for Authors
In the submitted paper, the authors compared the effects of two isoflavones, iridin and irigenin, with the most well-known ones—genistein, daidzein, and glycitein—in breast, prostate, and gastric cancer. At the beginning, the authors discuss flavonoids, presenting the structures of 12 classes of flavonoids, and mention the most important isoflavones. They describe in detail the cell signaling pathways and apoptosis mechanisms responsible for cancer development. After that, the anticancer effects of the selected isoflavones on the aforementioned cancers are discussed. In the DISCUSSION section, a similar text is repeated without providing a critical opinion, which is expected in a review manuscript.
In fact, there are many repeated sentences throughout this manuscript. For example, the following sentences appear in this paragraph: “Cancer is a worldwide issue and lots of research related to anticancer treatments such as isoflavones are actively conducted over the past few decades. Currently, Isoflavones are being studied for their anticancer effects because of its structural similarity of hormone estrogen which is called phytoestrogen. Isoflavones has led to numerous studies exploring their anti-cancer, anti-inflammation, and antioxidant properties. The most well-known isoflavones are genistein, daidzein, glycitein found in legumes. The most abundant isoflavone is genistein and daidzein. Glycitein is the third most abundant isoflavones, but majority of research are focused on genistein and daidzein. These compounds exhibit a range of biological activities due to their potential health benefits especially on breast, prostate and gastric cancer, “ these sentences are repeated multiple times throughout the manuscript. Please correct this repetition.
Generally, this work lacks a more concrete discussion, such as one based on the structural differences between the mentioned isoflavones and their distinct mechanisms of anticancer effects. Additionally, IC50 values (for cancer cell lines) for each isoflavone are not included, which would allow for a comparison of their anticancer potential. Furthermore, while there is a lot of basic information about cancer, apoptosis, and isoflavones, the essence of the work gets lost in the process. In other words, you cannot see the forest for the trees. Based on all these points, the reviewer’s opinion is that this manuscript should be rejected
Other remarks:
In the sentence (page 2, line 51): “The extrinsic pathway is regulated when death ligands bind to their death receptors, while the intrinsic pathway is activated when ROS level are high, or DNA is damaged” please explain better difference between the extrinsic and intrinsic pathways in cell apoptosis, for example what are “dead” ligand and “dead” receptor in order to readers who are not expert in this field can understand what you talk about.
Later in the manuscript the authors talk about the anticancer effect of isoflavones and note that it is induced by apoptosis. Please note at the beginning which type of apoptosis pathways is responsible for anticancer property of mentioned isoflavones.
Author Response
- In the submitted paper, the authors compared the effects of two isoflavones, iridin and irigenin, with the most well-known ones—genistein, daidzein, and glycitein—in breast, prostate, and gastric cancer. At the beginning, the authors discuss flavonoids, presenting the structures of 12 classes of flavonoids, and mention the most important isoflavones. They describe in detail the cell signaling pathways and apoptosis mechanisms responsible for cancer development. After that, the anticancer effects of the selected isoflavones on the aforementioned cancers are discussed. In the DISCUSSION section, a similar text is repeated without providing a critical opinion, which is expected in a review manuscript. In fact, there are many repeated sentences throughout this manuscript. For example, the following sentences appear in this paragraph: “Cancer is a worldwide issue and lots of research related to anticancer treatments such as isoflavones are actively conducted over the past few decades. Currently, Isoflavones are being studied for their anticancer effects because of its structural similarity of hormone estrogen which is called phytoestrogen. Isoflavones has led to numerous studies exploring their anti-cancer, anti-inflammation, and antioxidant properties. The most well-known isoflavones are genistein, daidzein, glycitein found in legumes. The most abundant isoflavone is genistein and daidzein. Glycitein is the third most abundant isoflavones, but majority of research are focused on genistein and daidzein. These compounds exhibit a range of biological activities due to their potential health benefits especially on breast, prostate and gastric cancer, “these sentences are repeated multiple times throughout the manuscript. Please correct this repetition.
- I have revised several repetitions.
- Generally, this work lacks a more concrete discussion, such as one based on the structural differences between the mentioned isoflavones and their distinct mechanisms of anticancer effects. Additionally, IC50 values (for cancer cell lines) for each isoflavone are not included, which would allow for a comparison of their anticancer potential. Furthermore, while there is a lot of basic information about cancer, apoptosis, and isoflavones, the essence of the work gets lost in the process. In other words, you cannot see the forest for the trees. Based on all these points, the reviewer’s opinion is that this manuscript should be rejected.
- I have added more information in my ‘Discussion’ section.
- My manuscript key point is that the metabolism of iridin and irigenin has not yet been fully revealed and additional researches are needed. I wrote this sentence at the ‘Introduction’ section and at the end of ‘Discussion’ section. I rearranged the order of my manuscript. I moved the original section ‘ Anticancer effects of various isoflavones’ to section 4, and shifted the original section 3 and 4 to section 2 and 3. Through this reorganization, I hope to clearly convey my key point.
- In the sentence (page 2, line 51): “The extrinsic pathway is regulated when death ligands bind to their death receptors, while the intrinsic pathway is activated when ROS level are high, or DNA is damaged” please explain better difference between the extrinsic and intrinsic pathways in cell apoptosis, for example what are “dead” ligand and “dead” receptor in order to readers who are not expert in this field can understand what you talk about.
- I added more detail information to explain the difference between extrinsic and intrinsic pathway.
- I added ‘death’ meaning at the ‘Apoptosis Pathway’ section.
- Later in the manuscript the authors talk about the anticancer effect of isoflavones and note that it is induced by apoptosis. Please note at the beginning which type of apoptosis pathways is responsible for anticancer property of mentioned isoflavones.
- In my manuscript, I mention that genistein and daidzein up-regulated BAX and down-regulated Bcl-2 at the end of intrinsic pathway explanation or iridin exhibited FAS at the end of extrinsic pathway, etc. Also, I mention how each isoflavones regulates signal pathway at the end of each part. I think these sentences seems to explain how each isoflavones regulates the apoptosis pathway and signaling pathway.

Round 3
Reviewer 1 Report (New Reviewer)
Comments and Suggestions for Authors
The manuscript is improved comparing with previous version so it can be accepted for publication.
This manuscript is a resubmission of an earlier submission. The following is a list of the peer review reports and author responses from that submission.
Round 1
Reviewer 1 Report
Comments and Suggestions for Authors
In the current review, the authors presented five isoflavones - genistein, daidzein, glycitein, irigenin, and iridin and their effects on three different cancers - breast cancer, prostate cancer, and gastric cancer.
Some suggestions:
1. Abstract, line 34 – you wrote about “the primary purpose is to summarize five isoflavones…”. Does the article have another purpose? 2. Introduction, lines 41-44 - Concerning the 2022 statistics, please add: - who made this statistics? -where was made? - it would be good to add a statistical study carried out in 2024 - please write only about the anticancer activity of isoflavones. Delete please anti-diabetic effects, anti-obesity, anti-osteoporosis…(lines 67-69, 75)3. Why did you included Subchapter 2: Apoptosis pathway, since no reference is made to how isoflavones are involved. Please complete.
4. Point 4. Flavonoid – You must write only about isoflavones. In addition, the title of the point 4.1 is: Structure and classification of flavonoid and you wrote also about biosynthesis and biologic activity.
5. point 2 of your article (after introduction) should be about isoflavones – structures, origin and a discussion concerning the relationship between structure and anticancer activity 6. Point 4.2.Isoflavones: delete please lines 205- 208. They are off topic.7. Point 5 (5.1-5.4.): Effect of isoflavones in various cancers:
The information regarding the origin and structure of isoflavones must be written at point 2.8. Delete please the following lines because they are out off topic:
-point 5.2 Daidzein and daidzin:lines 242-43
-point 5.3. Glycitein and glycitin – line 257
9. point 5.4. Irigenin and Iridin, lines 279-281: you wrote ” Another study proved that irigenin inactivated the MAPK signaling pathway in LPS-treated mice by regulating inflammatory response, cytokine production, and cell death in acute lung injury (Figure 6) [101]”. You wrote that you are following breast, prostate and gastric cancers.10. At point 5. Effect of isoflavones in various cancers - you must write also about in vivo studies- mice and clinical trials. First you must write about in vitro studies and then about in vivo studies.
11. Taking into account the fact that, unlike gastric cancer, breast and prostate cancers are hormone-dependent, please explain the differences in the action of isoflavones in the anticancer effect. Please add the explanation at discussion.Author Response
Thank you for your comments and suggestions. Based on your comments, I changed a few parts and highlighted them.
- Abstract, line 34 – you wrote about “the primary purpose is to summarize five isoflavones…”. Does the article have another purpose?
- Sorry for the confusion, I changed this part to make it clear
- Introduction, lines 41-44 - Concerning the 2022 statistics, please add:- who made this statistics? -where was made? - it would be good to add a statistical study carried out in 2024 - please write only about the anticancer activity of isoflavones. Delete please anti-diabetic effects, anti-obesity, anti-osteoporosis…(lines 67-69, 75)
- I added the ‘who’ part. The journal didn’t tell me the ‘where’, but the ‘who’ would make it clear. I tried to find the worldwide statistic 2024, but couldn’t find it. I deleted the part that you suggested
- Why did you included Subchapter 2: Apoptosis pathway, since no reference is made to how isoflavones are involved. Please complete.
- I added more information and references to complete it.
- Point 4. Flavonoid – You must write only about isoflavones. In addition, the title of the point 4.1 is: Structure and classification of flavonoid and you wrote also about biosynthesis and biologic activity.
- I deleted the ‘Flavonoid’ section and put it under the ‘Isoflavones structure and their role in human health’
- Point 2 of your article (after introduction) should be about isoflavones – structures, origin and a discussion concerning the relationship between structure and anticancer activity
- I moved it to come after the introduction and changed the title from ‘isoflavone’ to ‘isoflavones structure and their role in human health’
- Point 4.2.Isoflavones: delete please lines 205- 208. They are off topic
- I deleted this part
- Point 5 (5.1-5.4.): Effect of isoflavones in various cancers: The information regarding the origin and structure of isoflavones must be written at point 2.
- As I mention above, I moved this part to come after the introduction
- Delete please the following lines because they are out off topic: -point 5.2 Daidzein and daidzin:lines 242-43, point 5.3. Glycitein and glycitin – line 257
- I deleted this part.
- point 5.4. Irigenin and Iridin, lines 279-281: you wrote” Another study proved that irigenin inactivated the MAPK signaling pathway in LPS-treated mice by regulating inflammatory response, cytokine production, and cell death in acute lung injury (Figure 6) [101]”. You wrote that you are following breast, prostate and gastric cancers.
- I intended this part as an anti-inflammatory effect of irigenin, but it seemed to be unclear enough. I moved this part to ‘Discussion’ section
- At point 5. Effect of isoflavones in various cancers - you must write also about in vivo studies- mice and clinical trials. First you must write about in vitro studies and then about in vivo studies.
- I added each isoflavones clinical trial result at the ‘Discussion’ section
- Taking into account the fact that, unlike gastric cancer, breast and prostate cancers are hormone-dependent, please explain the differences in the action of isoflavones in the anticancer effect. Please add the explanation at discussion.
- I didn’t understand this comment clearly. The part where I wrote about the isoflavones effect on three different cancer seems to show the difference.
Reviewer 2 Report
Comments and Suggestions for Authors
The aim of the paper was to compare the anticancer activity of Iridin and Irigenin, lesser-known isoflavones, with activity of soy isoflavones.
In my opinion, the review is poor and does not fully cover the topic. There are many studies on the anticancer activity of daidzein and genistein, which are not mentioned and discussed here. Many review articles have been published on anticancer effects of isoflavones. For example, to discuss the anticancer effect of genistein, the authors relied on only five studies (see Table 1). The section on the anticancer effects of Iridin and Irigenin also does not fully cover the topic, even though the number of articles on this subject is much smaller than for soy isoflavones. See, for example, some papers that are not mentioned by the authors: doi.org/10.1186/s13765-020-00570-6; dx.doi.org/10.4314/tjpr.v20i7.6; doi.org/10.1038/srep37151.
Moreover, the authors should focus on the main topic. Several sections are unnecessary and too general, for example, the chapter on flavonoids in general.
There is also a lack of discussion on other mechanisms of action of isoflavones, such as anti-inflammatory and antioxidant effects, which also play a role in cancer development and anticancer activity.
There is no information about which databases were searched or what keywords were used.
Many pieces of information are unnecessarily repeated, np. Line 213: “Usually, natural form of isoflavones is glycosides” – the same information in line 214 (Usually, natural isoflavones are in the form of glycoside
Additionally, there are a few errors/inaccuracies in the text. For example:
“The most well-known glycoside form isoflavones are genistein, daidzein, and glycitein. Their aglycone form are genistin, daidzin, and glycitein…” – incorrect Genistin, daidzin, and glycitein are glycosides
Line 213: Usually, natural form of isoflavones is glycosides and are inactive – glycosides are also active
Isoflavones (Figure 3F), a subclass of flavonoid, are usually found in soy food or soy protein,” – isoflavones in soy protein?
Line 214: For anti-oxidation, aglycone has higher activity then glucosides.- unclear expression
„There are two main types on isoflavones which are called glycoside and aglycone.” – This is not a type but rather a form of occurrence.
Author Response
- 제 생각에, 이 리뷰는 형편없고 주제를 완전히 다루지 못했습니다. 다이드제인과 제니스테인의 항암 활성에 대한 연구는 많지만, 여기서 언급되거나 논의되지 않았습니다. 이소플라본의 항암 효과에 대한 리뷰 기사가 많이 게재되었습니다. 예를 들어, 제니스테인의 항암 효과를 논의하기 위해 저자는 5개의 연구에만 의존했습니다(표 1 참조). 이리딘과 이리게닌의 항암 효과에 대한 섹션도 이 주제에 대한 기사 수가 대두 이소플라본에 대한 것보다 훨씬 적음에도 불구하고 주제를 완전히 다루지 못했습니다. 예를 들어, 저자가 언급하지 않은 논문을 참조하세요: doi.org/10.1186/s13765-020-00570-6; dx.doi.org/10.4314/tjpr.v20i7.6; doi.org/10.1038/srep37151.
- 저는 이리게닌과 이리딘을 포함한 이소플라본의 항암 효과에 대해 논의하기 위해 더 많은 참고문헌과 정보를 추가했습니다.
- 게다가 저자는 주요 주제에 집중해야 합니다. 몇몇 섹션은 불필요하고 너무 일반적입니다. 예를 들어, 플라보노이드에 대한 장은 일반적입니다.
- '이소플라본 구조와 인체 건강에서의 역할' 섹션에 일부 내용을 삽입하여 '플라보노이드' 섹션을 제거했습니다.
- 또한 이소플라본의 다른 작용 기전, 즉 항염증 효과와 항산화 효과에 대한 논의도 있는데, 이러한 효과도 암 발병과 항암 활동에 중요한 역할을 합니다.
- 각 이소플라본의 항염 효과와 임상 실험 내용은 '토론' 섹션에 추가했습니다.
- 어떤 데이터베이스가 검색되었는지 또는 어떤 키워드가 사용되었는지에 대한 정보가 없습니다.
- 소개 부분의 마지막 부분에서 어떤 검색 엔진을 사용했고 어떤 키워드를 사용했는지 추가했습니다.
- 많은 정보가 불필요하게 반복됩니다. np. 213번째 줄: "일반적으로 이소플라본의 자연적 형태는 글리코사이드입니다." - 214번째 줄의 동일한 정보(일반적으로 자연적 이소플라본은 글리코사이드 형태입니다)
- 죄송합니다. 중복된 부분을 찾아서 삭제했습니다.
- 또한, 텍스트에 몇 가지 오류/부정확한 내용이 있습니다. 예를 들어:
- “가장 잘 알려진 글리코사이드 형태의 이소플라본은 제니스타인, 다이드제인, 글리시테인입니다. 이들의 아글리콘 형태는 제니스틴, 다이드진, 글리시테인입니다…” – 잘못된 예 제니스틴, 다이드진, 글리시테인은 글리코사이드입니다.
- 213번째 줄: 일반적으로 이소플라본의 자연적 형태는 글리코사이드이며 비활성입니다. 글리코사이드도 활성입니다. 플라보노이드의 하위 분류인 이소플라본(그림 3F)은 일반적으로 대두 식품이나 대두 단백질에서 발견됩니다. 대두 단백질에 이소플라본이 있습니까?
- 214번째 줄: 항산화 측면에서 아글리콘은 글루코사이드보다 활성이 더 높습니다.- 표현이 불분명합니다.
- 이소플라본에는 글리코사이드와 아글리콘이라는 두 가지 주요 유형이 있습니다.” – 이것은 유형이 아니라 발생 형태입니다.
- 오류가 있어서 죄송합니다. 해당 부분을 수정했습니다.

Round 2
Reviewer 1 Report
Comments and Suggestions for Authors In my opinion, unfortunately, the authors didn’t manage to improve the article enough.1. You wrote that you are following breast, prostate and gastric cancers. From discussions you must delete all the information concerning other types of cancer.
2. You wrote:
-lines 307-309, you wrote: “Genistein showed effective to pregnant woman before and during pregnancy. Genistein decreased gestational diatebes mellitus occurrence when pregnant woman took soy food mainly containing genistein [101].
-Lines 313-315: :Soy nuts that contain 10mg of glycitein lower the diastolic and systolic blood pressure in normotensive and hypertensive post-menopausal women [104].”
This information have nothing to do with the present article.
3. You must write also about in vivo studies- mice and clinical trials. First you must write about in vitro studies and then about in vivo studies.
4. Taking into account the fact that, unlike gastric cancer, breast and prostate cancers are hormone-dependent, please explain the differences in the action of isoflavones in the anticancer effect. Please add the explanation at discussion.
Author Response
Major revision round2
Reviewr 1
In my opinion, unfortunately, the authors didn’t manage to improve the article enough.
- You wrote that you are following breast, prostate and gastric cancers. From discussions you must delete all the information concerning other types of cancer.
- I deleted the part that are not related to breast, prostate, and gastric cancers.
- You wrote:
-lines 307-309, you wrote: “Genistein showed effective to pregnant woman before and during pregnancy. Genistein decreased gestational diatebes mellitus occurrence when pregnant woman took soy food mainly containing genistein [101].
-Lines 313-315: :Soy nuts that contain 10mg of glycitein lower the diastolic and systolic blood pressure in normotensive and hypertensive post-menopausal women [104].”
This information have nothing to do with the present article.
- I deleted these parts
- You must write also about in vivo studies- mice and clinical trials. First you must write about in vitro studies and then about in vivo studies.
- I added the in-vivo studies between the in-vitro and clinical trial paragraph at the ‘Discussion’ section
- Taking into account the fact that, unlike gastric cancer, breast and prostate cancers are hormone-dependent, please explain the differences in the action of isoflavones in the anticancer effect. Please add the explanation at discussion.
- I added the explanation at the ‘Discussion’ section

Reviewer 2 Report
Comments and Suggestions for Authors
The authors have made several improvements to their manuscript, but my main concerns remain.
The review is still inadequate and does not comprehensively cover the topic. For instance, the authors briefly describe the effect of genistein on breast cancer in just seven lines and cite only two references in Table 1. However, the topic is so extensive that warrants a full review article (see, for example, doi.org/10.3390/ijms25105556). For comparison, see also Table 1 in doi.org/10.3390/pr10020415, where the detailed activity of genistein against breast cancer is summarized.
The same comments applies to the description of other cancer types.
Furthermore, the title, "The Effect of Iridin and Irigenin in Cancers: Comparison with Well-Known Isoflavones," suggests a comparative analysis. However, no substantial comparisons are provided, whether in mechanisms of action, dosage, or efficacy.
Author Response
Major revision round2
Reviewr 2
The authors have made several improvements to their manuscript, but my main concerns remain.
The review is still inadequate and does not comprehensively cover the topic. For instance, the authors briefly describe the effect of genistein on breast cancer in just seven lines and cite only two references in Table 1. However, the topic is so extensive that warrants a full review article (see, for example, doi.org/10.3390/ijms25105556). For comparison, see also Table 1 in doi.org/10.3390/pr10020415, where the detailed activity of genistein against breast cancer is summarized.
The same comments applies to the description of other cancer types.
- I added more references on each isoflavones effect in the text and table 1. Also, I divided the ‘Table 1’ to make it clear to see the comparison of each effect of isoflavones
Furthermore, the title, "The Effect of Iridin and Irigenin in Cancers: Comparison with Well-Known Isoflavones," suggests a comparative analysis. However, no substantial comparisons are provided, whether in mechanisms of action, dosage, or efficacy.
- I added comparison of hormone-dependent and non-hormone dependent, in-vivo studies, and clinical studies at the ‘Discussion’ section. Also, Table 1 seems to show the dosage and effect of isoflavones in each cancer.

Round 3
Reviewer 1 Report
Comments and Suggestions for Authors In my opinion, unfortunately, the authors didn’t manage to improve the article enough.1. You wrote that you are following breast, prostate and gastric cancers. From discussions you
3. You must write also about in vivo studies- mice and clinical trials. First you must write about in vitro studies and then about in vivo studies.
4. Taking into account the fact that, unlike gastric cancer, breast and prostate cancers are hormone-dependent, please explain the differences in the action of isoflavones in the anticancer effect. Please add the explanation at discussion.